# Easily Attach/Detach Reattachable EEG Headset with Candle-like Microneedle Electrodes

**DOI:** 10.3390/mi14020400

**Published:** 2023-02-06

**Authors:** Takumi Kawana, Yuki Zemba, Ryo Ichikawa, Norihisa Miki

**Affiliations:** Department of Mechanical Engineering, Keio University, Yokohama 223-8522, Kanagawa, Japan

**Keywords:** electroencephalography, headset, long-term measurement, spontaneous potential, mental state, dry electrodes, micro-needle

## Abstract

To expand the applications of the electroencephalogram (EEG), long-term measurement, a short installation time, and little stress on the participants are needed. In this study, we designed, fabricated, and evaluated an EEG headset with three candle-like microneedle electrodes (CMEs). The user is able to detach and reattach the electrodes, enabling long-term measurement with little stress. The design of the CMEs was experimentally determined by considering the skin-to-electrode impedance and user comfort. An EEG was successfully measured from areas with a high hair density without any preparation. The installation time was shorter than 60 s and the electrodes could be detached and reattached. The headset was designed such that the discomfort caused by its ear pads was higher than that caused by the electrodes. In 1 h experiments, the participants did not feel pain and the detachment of the CMEs was found to improve the comfort level of the participants in most cases. A successful demonstration of the long-term measurement of EEGs while watching a whole movie verified that the developed EEG headset with CMEs is applicable for EEG measurement in a variety of applications.

## 1. Introduction

Electroencephalography (EEG) is the measurement of electrical activity on the scalp by electrodes. It reflects a part of the neural activity of the brain and has been used to diagnose epilepsy and other brain disorders [1,2,3,4,5] as well as aid the rehabilitation of patients with paralysis [6,7,8]. Since EEG is mostly non-invasive, its potential applications in non-medical fields, such as estimation of emotions and evaluation of preferences, have been explored [9,10,11,12,13,14,15]. 

Such EEG applications involve preparation, measurement, signal processing, and analyses. The present study focuses on preparation and measurement. Preparation has three steps: skin treatment, hair handling, and fixation of the electrodes. When flat disk electrodes are used, the high-impedance stratum corneum at the measurement site is removed and conductive glue is applied to secure a conductive contact between each electrode and the skin [16,17]. These processes are time-consuming and the sticky glue annoys participants. Dry electrodes, which have pin-shaped electrodes and can be pressed to the skin with a large pressure to reduce the impedance even with the presence of the stratum corneum, have been proposed [18,19,20,21,22,23,24,25]. However, dry electrodes apply a relatively strong pressure to the skin, which is undesirable, particularly for long-term measurements. 

For both flat disk and dry electrodes, avoiding hair is a challenge [26,27]. For the 10–20 measurement system, only FP1 and FP2, which are on the forehead, are not influenced by hairs. Our group previously proposed and demonstrated candle-like micro-needle electrodes (CMEs, see Figure 1) [28,29,30]. CMEs can avoid hair and their sharp tips penetrate the stratum corneum, which ensures sufficiently low contact impedance between the electrodes and the skin. The pressure they apply to the skin is lower than that for other dry electrodes and they do not cause pain. CMEs were experimentally verified to have as good a performance as conventional flat electrodes [28] and can capture both spontaneous potential and event-related potential [29].

Fixation of the electrodes onto the scalp is another challenge. Several mechanisms that stably fix electrodes to the skin have been developed and commercially available. They are divided into those that use a net, a cap, and a compliant mechanism (summarized in [31]). Net- and cap-type EEG systems use stretchable fabrics to distribute the pressure over a large area, which enable a large number of electrodes to be in contact with the skin. However, the pressure applied to each electrode is limited and, thus, a gel or saline solution is necessary to ensure good electrical contact with the scalp. Headsets that use a compliant mechanism provide sufficient pressure to the electrodes to achieve a sufficiently low electrode–skin impedance.

During EEG measurement, participants are likely to feel discomfort at the measurement sites. This discomfort can become significant in long-term measurement [32,33,34]. Continuous pressure at the same location is not physiologically favored. However, EEG electrodes are designed to maintain continuous and stable contact with the skin in many studies.

In this work, we propose an EEG measurement system that overcomes the challenges associated with EEG preparation and measurement. As shown in Figure 2, the system is a headset (a pair of headphones system with CMEs). The design is based on the following three ideas:1.Since CMEs do not require any skin preparation or hair avoidance, and headphones are commonly used, the proposed EEG headset can be comfortably worn by users.2.Headphones are commonly worn for hours while pressure is applied by ear pads. If the pressure applied by the EEG electrodes to the skin is lower than that applied by the ear pads to the ear, the EEG measurement system should be accepted by users. The pressure applied by the electrodes is adjusted by springs on the headset such that the induced stimuli are sufficiently small while ensuring good contact between the electrodes and the skin.3.Headphones are often moved by the user when they feel discomfort caused by the ear pads. Therefore, the electrodes need to be detachable and not fixed to the skin. This is enabled by the optimal design of the CMEs. EEG measurement is interrupted when the electrodes are detached from the scalp. Shortly after the headset returns to its original position, EEG measurement resumes since the CMEs are reattached immediately and automatically while avoiding hair and penetrating the stratum corneum.

The design of the CMEs and EEG headset was determined based on experiments. The preparation required for using the designed EEG headset was experimentally evaluated. EEG measurement for more than 1 h was conducted to verify the effectiveness of the proposed EEG headset. All the experiments were approved by the research ethics committee of the Faculty of Science and Technology, Keio University (2020-33).

## 2. Design of CMEs

### 2.1. Design and Fabrication

The CMEs were designed to enhance hair avoidance, decrease electrode–skin impedance, and optimize user friendliness. Here, we define the space S, which holds the avoided hair, as follows,
(1)S=L×w,
where  w is the pillar spacing and L is the pillar length (see Figure 3). S needs to be sufficiently large for the electrodes to automatically avoid hair. L and the aspect ratio of the electrodes are limited by the fabrication capability. A larger w leads to a smaller number of pillars and thus fewer contact areas, which leads to higher impedance. In addition, fewer pillars may cause users pain. The optimal values of L and w were experimentally determined. In the experiments, we set the electrode dimensions to 10 mm × 10 mm given the spatial resolution of an EEG. A pillar diameter of 0.4 mm and a sharp tip length of 0.2 mm were verified to be appropriate in our prior work [28]. w was determined based on the number of electrode pillars.

The fabrication processes for the CMEs is detailed in our prior work [28]. Briefly, a brass mold in the shape of the CMEs was micro-machined using a five-axis control vertical machining center. Then, a second mold, made of polydimethyl siloxane (PDMS; SYLGARD 184 W/C, Dow Corning Toray Co., Chiyoda, Japan), was formed using the brass mold. Subsequently, micro-needle arrays made of the negative photoresist SU-8 (SU-8 10, Kayaku Microchem, Tokyo, Japan) were formed using the PDMS mold. A 0.1 μm-thick silver film was deposited onto the SU-8 micro-needle arrays in order to provide electrical conductivity. Although a large L is preferable to have a large space S, taking the fabrication accuracy into consideration, L was designed to be 2.8 mm, which resulted in an electrode height of 3.0 mm and a tip length of 0.2 mm. Electrodes with 36, 49, 64, 81, 100 and 144 pillars were fabricated, for which w was 1.4, 1.1, 0.89, 0.73, 0.60 and 0.43 mm, respectively (see Table 1). For the electrode with 144 pillars, the length L was limited to 0.8 mm due to the fabrication capacity. The needle sizes and shapes at the pillar tips were the same for all electrodes. The CME with 36 pillars is shown in Figure 1.

### 2.2. Experiments

The skin-to-electrode impedance reflects the electrical characteristics of the interface between the electrode and the skin. A low and stable skin-to-electrode impedance leads to a low signal-to-noise ratio and accurate EEG measurements. For dry electrodes, a larger contact area results in a lower impedance [35]. The impedance needs to be below 300 kΩ to perform a measurement with a wireless biometric device (Polymate Mini AP108, Miyuki Giken, Tokyo, Japan). 

The impedance measurement was conducted on seven participants (six males and one female in their 20s). The measurement site was the top of the head (Cz in the 10–20 system). A previously reported CME (144 pillars with L of 0.8 mm) and five newly fabricated CMEs (36, 49, 64, 81 and 100 pillars with L of 2.8 mm) were tested. The electrode was attached to a copper holder (for details of the design, see Appendix A) and the electrode holder was fixed to a digital force gauge (DST-500N, Imada Co., Ltd., Toyohashi, Japan), as shown in Figure 4a. Each electrode was pressed against the top of the head with a force of 1, 2, 3, 4 and 5 N, respectively, while the impedance was measured. The comfort level of the participants during measurement was evaluated via a questionnaire evaluation with a 4-point scale for each electrode and force (1: comfort, 2: pressure but no pain. 3: slight pain, 4: strong pain). The electrodes were connected to a wireless biometric device (Polymate Mini AP108, Miyuki Giken) using an electrode cord (AP-C131-015, Miyuki Giken). Kendall^TM^ electrodes (H124SG, CardinalHealth, Japan) were attached at the right mastoid process (neutral electrode) and the left mastoid process (reference electrode). The electrodes had conductive and adhesive hydrogel and could be adhered to the skin firmly while also being able to be removed without leaving any adhesives on the skin. They were then connected to the biometric device via electrode cords (AP-C132-015, Miyuki Giken, Japan). The tested electrodes and Kendall^TM^ electrodes served as the active electrodes.

The obtained impedance is summarized in Figure 4b. All of the newly fabricated electrodes were found to have a lower impedance than that of the previously reported CME with 144 pillars. This indicates that the CMEs with a sufficiently large space S successfully avoided hair and had a low skin-to-electrode impedance. The electrode with 64 needles and a pillar height of 2.8 mm demonstrated the lowest impedance. 

Figure 4c shows the results of the questionnaires about comfort for each electrode. Participants were more likely to feel pain with fewer needles on the electrode. The electrode with 64 needles was found to cause no pain for any participant at force of 1 N while maintaining a sufficiently small impedance. For 75% of the participants, measurements could be performed without causing pain when the force was smaller than 3 N.

Based on these results, the electrodes with 64 needles is considered to be appropriate as it efficiently avoids hair and has a low skin-to-electrode impedance. A force of 1 N applied by the electrode to the skin was found to be a good target for EEG measurement with a low skin-to-electrode impedance (<~100 kΩ) and no pain.

## 3. EEG-Headset

### 3.1. Overall Design

A conceptual sketch of the proposed headset is shown in Figure 2a. The EEG headset consists of a pair of commercially available headphones (ATH-S100 BBL, audio-technica, Tokyo, Japan) with attachment parts for the CMEs. The headset has three electrodes, one each for the left and right temporal regions (T3 and T4) and the parietal region (Cz). T3 and T4 have been reported to show large β-waves, which indicate tension and agitation [10,36,37]. Cz is frequently used to detect event-related potentials [38]. 

As shown in Figure 5a, the attachment parts are composed of springs and the 3D-printed mount parts made of acrylic resin. The CMEs are set in electrode holders made of copper, which are connected to a biosignal acquisition system (Polymate Mini, Miyuki-Giken, Japan) with lead wires. The headset is worn as a conventional pair of headphones. As mentioned above, since headphones are commonly used for long periods of time, when users perceive that the pressure applied by the CMEs is lower than that applied by the ear pads, the proposed EEG measurement system is considered to be accepted by users. The springs and mount parts were designed to satisfy this requirement.

### 3.2. Electrode Attachment Parts

#### 3.2.1. Temporal Regions (T3 and T4)

Springs are used to compliantly connect the electrodes to the headset. They need to be selected to ensure stable contact of the electrodes to the skin while keeping the perceived pressure applied by the electrodes lower than that applied by the ear pads. Figure 5b shows a conceptual diagram of the electrode attachment parts, where $l_{0}$ is the natural length of the spring, $d_{ele}$ and $d_{m}$ are the thicknesses of the electrode part and the mount part, respectively, $l_{head-hb}$ is the distance between the scalp and the inner wall of the headband when the headset is set, and s is the deformation of the spring. The relationship among the parameters is expressed as:(2)$$s+l_{head-hb}=l_{0}+d_{m}+d_{ele}.$$

The pressing force applied by the electrode $F_{ele}$ can be described using the spring constant k as,
(3)$$F_{ele}=ks=k\left(l_{0}+d_{m}+d_{ele}-l_{head-hb}\right)$$

$d_{ele}$ was set to 13.2 mm. $l_{head-hp}$ is determined by the deformation of the headband and thus depends on the shape and size of the participants’ heads. We experimentally investigated the variation of $l_{head-hp}$ among 11 participants (males in their 20s). The participants were requested to put on the headset by themselves and $l_{head-hb}$ was measured. $l_{head-hb}$ was found to be 27.1 ± 3.5 mm (a variation of 13%). The deformation or opening of the headband ranged from 190.0 to 219.8 mm and the force applied by the ear pads $F_{hs}$ was found to be 3.1 ± 0.2 N on average (see Table 2 for the full data). 

Next, we attempted to experimentally determine the spring stiffness that provides the appropriate force to the electrodes, i.e., a lower perceived pressure than that applied by the ear pads and sufficient force to ensure low skin-to-electrode impedance. The ratio of the force $F_{ele}$ to the force applied by the ear pads, or the headset, $F_{hs}\;$ was experimentally investigated. It is expressed as
(4)$$\frac{F_{ele}}{F_{hs}}=\frac{k\left(l_{0}+d_{m}+d_{ele}-l_{head-hb}\right)}{F_{hs}}.$$

Ten springs with spring constants of 0.255 to 3.03 N/mm were tested. Five participants (B, C, D, F, and H in Table 2, males in their 20s) participated in the experiments. Considering the size of the head, $\left(l_{0}+d_{m}+d_{ele}\right)$ in Equation (4) was set to 31.4 and 34.9 mm for three participants (B, D, F) and two participants (C, H), respectively. 

The discomfort ratio between the CMEs and the ear pads was assessed using the visual analog scale shown in Figure 6a. The participants were requested to plot the discomfort ratio on the scale after wearing the headset for 30 s. The length of the entire scale presented to the participant was $x_{0}$ and the length from the end point of the ear pad side to the plot position was $x_{hs}$. The discomfort level of the headset $I_{hs}$ is expressed as
(5)$$I_{hs}=1-\frac{x_{hs}}{x_{0}},$$

We would like to design the electrode attachment part such that $I_{hs}$ is larger than 0.5 (i.e., the participants perceived the pressure applied by the ear pads to be higher than that applied by the electrodes). In the first two trials, springs with the largest and smallest spring constants were tested. The remaining eight springs were then randomly tested. The spring constants for the tested springs are summarized in Table 3.

The results are shown in Figure 6b. $F_{ele}/F_{hs}$ was calculated using the average values of $l_{head-hb}$ and $F_{hs}$. For all subjects, the headset discomfort level $I_{hs}$ tended to decrease as $F_{ele}/F_{hs}$ increased, i.e., as the force applied by the electrodes increased. When $0<F_{ele}/F_{hs}\leq 1$, $I_{hs}$ was higher than 0.5 for all five participants, which indicates that the participants perceived the pressure applied by the electrodes to cause less discomfort than that applied by the ear pads. Springs that satified $0<F_{ele}/F_{hs}\leq 1$ were thus selected.

As found in the experiments, $l_{head-hb}$ varied among the participants (see Table 2). In order to minimize the effect of variation, a small spring constant k is preferable. The deformation of the spring, which is ($l_{0}+d_{m}+d_{ele}-l_{head-hb})$, also needs to tolerate the variation of $l_{head-hb}$ (±3.5 mm). The amount of deformation thus needs to be larger than 7 mm. Another design constraint, which was found in the experiment, is that buckling of the springs needs to be avoided. The buckling of the springs takes place when the electrodes contact the skin at some angles. To avoid buckling, the participant must conduct the attachment process carefully and possibly multiple times, which makes the setup time long. The mount part is designed such that the electrodes approach the head perpendicularly. In order to further prevent buckling, the aspect ratio of the spring was designed to be sufficiently small (i.e., the length is short and the outer diameter of the spring is large). The outer diameter of the coil springs needs to fit the electrode part and the mount part, which are slightly larger than the electrodes (10 mm × 10 mm). Due to these constraints, we decided to use the coil-type spring UY12-15 (Misumi Group Inc., Tokyo, Japan), which has a spring constant of 0.2 N/mm, a length $l_{0}$ of 15 mm, a maximum allowable deflection of 11.25 mm, and an outer diameter of 12 mm.

In order to satisfy $F_{ele}/F_{hs}<1$, we prepared attachment parts with three sizes (large, medium, and small) to further compensate for the variation in $l_{head-hs}$. For participants with $l_{head-hs}<24.5\;\mathrm{mm}$, $d_{m}$ was 2 mm (size S); for participants with $24.5\;\mathrm{mm}\leq l_{head-hs}\leq 28.5\;\mathrm{mm}$, $d_{m}=6\;\mathrm{mm}$ (size M); for participants with $28.5\;\mathrm{mm}\leq l_{head-hs}$, $d_{m}=10\;\mathrm{mm}$ (size L). For $l_{head-hs}$, the minimum value was 23.2 mm (participant D) and the maximum value was 33.5 (participant F, see Table 2). The resulting $F_{ele}$ was $1.14<F_{ele}\leq 1.4$, $1.14\leq F_{ele}\leq 1.94$, and $0.94\leq F_{ele}<2.0$ for sizes S, M, and L, respectively. Since $F_{ele}$ was larger than or close to 1.0 N, the impedance was considered to be sufficiently low (see Figure 4b) while $0<F_{ele}/F_{hs}\leq 1$ was satisfied. According to the experiments with only electrodes (i.e., no headset), as shown in Figure 4, these ranges of $F_{ele}$ resulted in pain levels smaller than 2; i.e., the participants felt the pressure but no pain, which was considered to be acceptable.

#### 3.2.2. Parietal Region (Cz)

The CME on the top of the head contacts the scalp while wearing the headphones. The attachment part for the top of the head has a spring. In addition, guide rods are designed in the attachment part for the top of the head to prevent the spring from buckling.

Since the CME in the parietal region is far from the ear pads, the pressure applied by the CME to the skin may not be masked by that applied by the ear pads. The discomfort caused by the electrodes in the parietal region was evaluated independently from that in the temporal region. We investigated the effect of the spring and then designed and evaluated a mechanism that reduces the discomfort caused by the CME.

We tested the four springs shown in Table 4. The CME was attached to only the top of the headset (parietal region). Five participants (five males in their 20s) were asked to assess the discomfort level from 0 (no discomfort) to 10 (strong discomfort). Before the experiment, the participants wore the headset with the CME but without the spring; the discomfort level was 5 (used as a reference). The results for the four springs are summarized in Figure 7a. A one-group *t*-test with a significance level of α = 0.05 was conducted. The null hypothesis was that the mean value of the questionnaire results for each spring is 5. The *p*-value for all springs was higher than 0.05 and thus the null hypothesis was not rejected. Since the discomfort levels for the springs did not have a significant difference, we selected spring 3, which provides a maximum load of 1.2 N at the top of the head, which is sufficiently large to maintain a low skin–electrode impedance.

To further reduce the discomfort of the electrodes on the top of the head, a headband cushion (HD545TL, TDITD Inc, Tokyo, Japan) was placed in the vicinity of the electrode. The thickness of this cushion was 18 mm, which is about the same as the thickness of the attachment part for the temporal region. Five participants (five males in their 20s) assessed the discomfort level from 0 (no discomfort) to 10 (strong discomfort). The participants first wore the headset without the cushion as a reference (the discomfort level was set to 5). The results are shown in Figure 7b. A one-group *t*-test with a significance level of α = 0.05 was conducted for the results obtained with the cushion. The null hypothesis was that the mean value of the questionnaire results was 5. The null hypothesis was rejected; there was a significant difference in the discomfort level of the participants caused by the electrodes with and without the cushion. Therefore, the cushion was set on the headset.

### 3.3. EEG Measurement with Developed EEG Headset

The developed EEG headset with CMEs has the following advantages over conventional EEG measurement systems.

1.Short installation time; no skin preparation is necessary before the measurement. Since the headset can be worn like a pair of headphones, the participants can quickly put it on (the CMEs automatically avoid hair). The skin–electrode impedance is sufficiently low when the needles of the CMEs penetrate the stratum corneum.2.Discomfort caused by the CMEs is acceptable since it is designed to be smaller than that caused by the ear pads of the headset. This is crucial for long-term measurements.3.The proposed headset allows the participants to detach and re-attach the headset, which enables long-term measurement. Conventional EEG electrodes are designed to be in stable contact with the skin throughout an experiment, whereas the proposed CMEs can be detached from the skin. Shortly after they are re-attached to the skin, the EEG measurement resumes since the CMEs automatically avoid hair and the needles penetrate the stratum corneum without any skin preparation. When the participants feel discomfort caused by the ear pads or electrodes, they can detach them and then reattach them when the discomfort is gone. Since the spatial resolution of EEG measurement is considered to be around 1 cm, the measurement site after detachment does not need to exactly match that before detachment. This approach enables participants to continue the EEG measurement in comfort.

The developed EEG headset was experimentally evaluated with one and three CMEs (that with two CMEs was verified in the previous section). The EEG headset with three CMEs at Cz, T3, and T4 was used. Kendall^TM^ electrodes were used for the reference electrode on the left mastoid process and the neutral electrode on the right mastoid process. Polymate Mini AP108 (Miyuki-Giken) was used to collect, amplify, and transmit the EEG signals from all the electrodes while the impedance was monitored.

#### 3.3.1. Wearing of EEG Headset

The time required for the measurement to start was measured for five participants (males in their 20s; A, B, C, D and E; these letters do not correspond to the participants in Table 2). The setup was considered to be complete when the impedance of each CME was less than 300 kΩ (when Polymate Mini started measuring the EEG). After it was confirmed that none of the wires were tangled, the participants put on the headset with or without help from the experimenter. In the setup, the headset was opened widely, the position of the CME at Cz was adjusted, and then the headset was closed to let the CMEs at T3 and T4 contact the skin. Finally, the reference and neutral electrodes were attached. This setup process was conducted five times for each participant. For comparison, the setup time of a disc electrode (NE-113A, NIHON KOHDEN, Tokyo, Japan) at Cz was also measured. 

The setup time for the EEG headset and the disk electrode for each participant and the skin-to-electrode impedance of all electrodes are shown in Figure 8. The setup time for the EEG headset is much shorter (27.4 ± 7.6 s and 15.0 ± 3.9 s with and without help from the experimenter, respectively) than that for the disk electrode (205 ± 19 s). Interestingly, the setup time was shorter when the participants put on the headset by themselves.

It has been reported that the setup time for a commercially available wearable EEG headset is a few minutes [39]. In contrast, the proposed EEG headset can be set up in less than 1 min. The disk electrode requires the hair to be brushed, the stratum corneum to be removed, and electrolyte paste to be applied.

#### 3.3.2. Detachment and Reattachment of CME

We conducted experiments to determine the time required to resume measurement after the participant detaches and reattaches the electrodes. Five participants (males in their 20s; A, B, C, D and E; these letters do not correspond to the participants in the previous experiments) wore the headset and then opened the headband to separate the electrodes from the skin (detachment). Then, they let the headband close and the electrodes contact the scalp. The time required to resume measurement, during which the impedance returned to below 300 kΩ, was measured. Each participant conducted the experiment five times. As shown in Figure 9, the time required to resume measurement was below 10 s, except for T4 for participant B. This short period of time required to resume measurement is considered to be acceptable for long-term experiments. 

#### 3.3.3. Long-Term Measurement

We conducted a 1 h EEG measurement. Five participants (males in their 20s; A, B, C, D and E; these letters do not correspond to the participants in the previous experiments) participated in the experiments. Before the experiments, the participants were informed that they could detach and reattach the CMEs during the experiments. They reported the discomfort level using the visual analogue scale (see Figure 6a) every 5 min. The ratio of discomfort between the headset and the electrodes $I_{hs}$ was calculated using Equation (5). The skin-to-electrode impedance was measured every 5 min to verify good contact between the electrodes and the skin. The results are shown in Figure 10.

The proposed EEG headset enables comfortable long-term measurement for 1 h. As shown in Figure 10, 50 out of 65 points (13 values × 5 participants) were above 0.5 (i.e., the participants considered that the discomfort caused by the ear pads dominated that caused by the CMEs). Participant D has an $I_{hs}$ value higher than 0.7 throughout the experiment. Participants B and C have low $I_{hs}$ values at the beginning of the experiments and then $I_{hs}$ values higher than 0.5 after 45 and 10 min. Participant A has a relatively high $I_{hs}$ value until 50 min; the value rapidly decreases in the final 10 min. $I_{hs}$ for participant I fluctuates.

The detachment/reattachment time for the headset is also shown in Figure 10. Participant E has a low $I_{hs}$ value at 25, 40, and 55 min. After detachment/reattachment was conducted, $I_{hs}$ increased. This verifies that detachment/reattachment of the headset enables long-term measurement. Participant B has a higher $I_{hs}$ value after 40 and 50 min due to detachment/reattachment. Though Participant A detached and attached the headset, his $I_{hs}$ did not recover.

#### 3.3.4. Monitoring of Mental State of Participants during Movie Watching

The EEG of three participants (males in their 20s; A, B and C; these letters do not correspond to the participants in the previous experiments) were acquired from Cz, T3, and T4 using the developed headset while the participants watched a movie for 90 min. We selected the movie “Paranormal Activity” (Oren Peli, 2007) since the scary parts mainly take place at night (and thus can be easily labeled) and the scary and non-scary scenes alternate throughout the movie. 

Note that this experiment was intended to demonstrate the long-term measurement of the EEGs. Although we attempted to correlate the acquired EEG to the mental state of the participants, the obtained results were preliminary and inconclusive.

The protocol of the analysis was as follows. A notch filter at 50 Hz, which is the mains frequency in the Tokyo area, and a bandpass filter from 3 to 30 Hz were applied to the acquired data. The data during detachment/reattachment were considered to be lost data and excluded from the analysis. 

The data were divided into 10,000 ms segments with an overlap of 50%. When the maximum amplitude of the obtained EEG was larger than 100 µV, the segment was considered to contain significant noise and excluded from the following analysis. The power spectral density (PSD) was estimated using the Welch method with a window size of 2048 and an overlap of 50%. The PSD with respect to time was obtained in the experiments.

The PSD in the high-β band (18–29.75 Hz) was investigated. It has been reported that muscle artifacts overlap high-βwaves [40]. The segments that had a PSD in the high-β band higher than the threshold, which was determined to be the top 10% of the PSD in the band, were excluded as lost data. When the PSD in the high-β band was too small, i.e., the bottom 5%, the measurement was unsuccessful. Such segments were also excluded as lost data. Table 5 shows the ratio of the lost data to all acquired data for each measurement site and participant. Large amounts of lost data were found for participant C at T4; however, the segments with lost data were found to be scattered and did not hinder the monitoring of the mental state. In other regions, the lost data were limited to 8% to 26%.

The ratio of PSD in the high-β band to that in the α-band, high-β/α, was considered to represent fear and tension [36,41,42]. Given that the participants were watching a horror movie, high-β/α can be a good measure for investigating their mental state. However, it is unknown how high-β/α quantitatively corresponds to fear. We describe the processes used in our work to estimate fear using high-β/α in the following section. 

The fear state is considered to last for a while. The PSD of high-β/α was calculated for each segment (10,000 ms with an overlap of 50%). In this work, we decided to evaluate fear as a binary value (i.e., fear/non-fear). We thus needed to determine a threshold. Figure 11 shows the cumulative percentile of the PSD of high-β/α for participant A. As shown, the top 10% of the PSD has the largest values. Therefore, we set the top 10% as the threshold.

Figure 12 and Figure 13 show the normalized PSD for high-β/α measured at Cz, T4, and T3 with respect to time, and the time when the percentile rank was top 10% (i.e., the participant was presumed to feel fear) for each participant. Figure 12 and Figure 13 summarize the results with respect to each participant and each measurement site, respectively. The number of fear moments is the same for all measurement sites (top 10% of all samples). The distribution of fear moments is shown in the figures. It should be noted that each dot corresponds to 60 s for better visualization in the figures. Figure 14a shows the time when the PSD for high-β/α was in the top 10% (i.e., fear time) for all participants at all measurement sites. In the movie, “Paranormal Activities”, scary parts take place during the night. The night scenes are highlighted in the figure. We determined the average number of measurement sites, 9 at maximum (3 participants × 3 measurement sites) in the top 10% of the PSD per segment during the day and the night in the movie.

The following findings were obtained.

1.EEGs were successfully acquired from the three measurement sites for all three participants during the whole movie despite some periods of no data due to detachment of the CMEs and artifacts caused by body movement.2.EEGs from Cz, T4, and T3 show some correspondence (inconclusive). For example, Figure 12a shows that all three EEGs had a top 10% PSD for high-β/α around the times of 800, 4000, 4800, and 5000 s.3.It is difficult to determine which measurement site is most suitable for obtaining EEGs to assess fear. The participants did not have identical mental states. However, since they watched the same horror movie at the same time, we expected the EEGs from the three participants to show some similarity. In Figure 13, some similarity can be seen, but the results are inconclusive.4.Figure 14a shows some regions with dense dots. As shown in Figure 14b, more fear moments were detected during the night scenes. For participant A, the difference was small, whereas for participants B and C, it was large. A statistically significant difference was not found (*p* = 0.11 when the null hypothesis, namely, that there is no difference between the day and night scenes, is considered). Note that experiments with more participants could show a statistically significant difference.5.Some scary parts in the movie take place during the day. We may obtain more conclusive results if the times in the movie are more accurately classified as scary and non-scary.

As described above, these experiments were conducted to demonstrate a long-term and practical EEG measurement using the proposed EEG headset. Assessment of fear and tension using the PSD for high-β/α will be reported in the near future.

## 4. Conclusions

In this study, we designed, developed, and evaluated an EEG headset with three CMEs, which allows the participants to detach and reattach the electrodes and thus enables long-term measurement with little stress on the participants. 

The CMEs with a pillar length of 3 mm and 64 pillars were able to measure EEGs in areas with large hair density without any preparation, which enables a short installation time and detachment/reattachment. 

When the perceived pressure applied by the electrodes was lower than that applied by the ear pads, the participants stated that the discomfort caused by the ear pads of the headset dominated that caused by the electrodes. When designed so, the EEG headset should be accepted by users like conventional headphones. The installation time is no longer than 60 s, which is shorter than that for other electrodes. It was experimentally verified that the participants can wear the headset for over 1 h without feeling pain. After detachment and reattachment of the electrodes, it took only 10 s to resume the measurement. The detachment was found to improve the comfort level of the users in most cases. Successful demonstration of the long-term measurement of EEGs during a whole movie verified that the developed EEG headset with CMEs is readily applicable for EEG measurement in a variety of applications.

## Figures and Tables

**Figure 1 micromachines-14-00400-f001:**
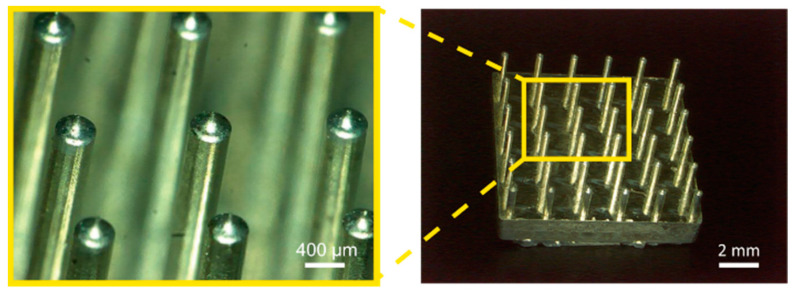
Images of developed CMEs. The pillar structures automatically avoid hair and the sharp tips penetrate the high-impedance stratum corneum.

**Figure 2 micromachines-14-00400-f002:**
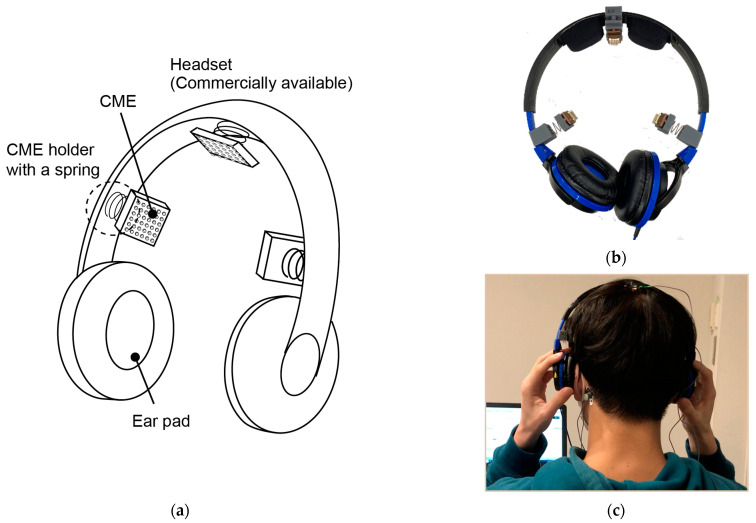
Proposed EEG headset. (**a**) Conceptual sketch. Three CMEs are attached to the commercially available headset with compliant springs. The springs are selected such that the perceived pressure applied by the CMEs is lower than that applied by the ear pads. Photographs of (**b**) proposed EEG headset and (**c**) user wearing EEG headset. The reference and neutral electrodes are attached to the back of the neck.

**Figure 3 micromachines-14-00400-f003:**
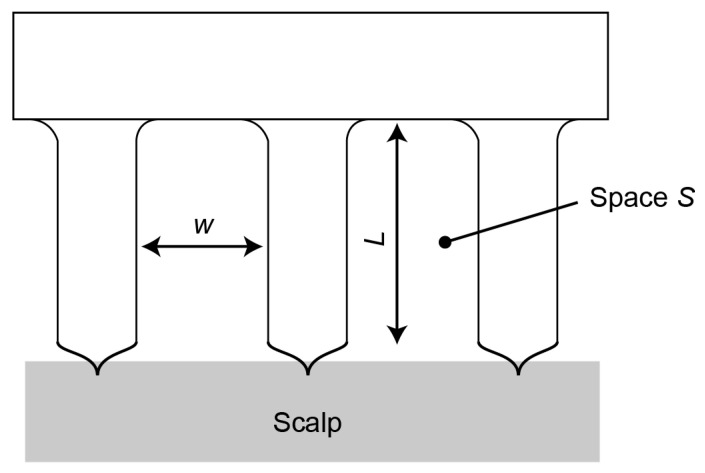
Design of CMEs. All CMEs have a square shape with a width of 10 mm. The number of pillars determines the gap width w (see Table 1). The diameter of the pillars is 0.4 mm.

**Figure 4 micromachines-14-00400-f004:**
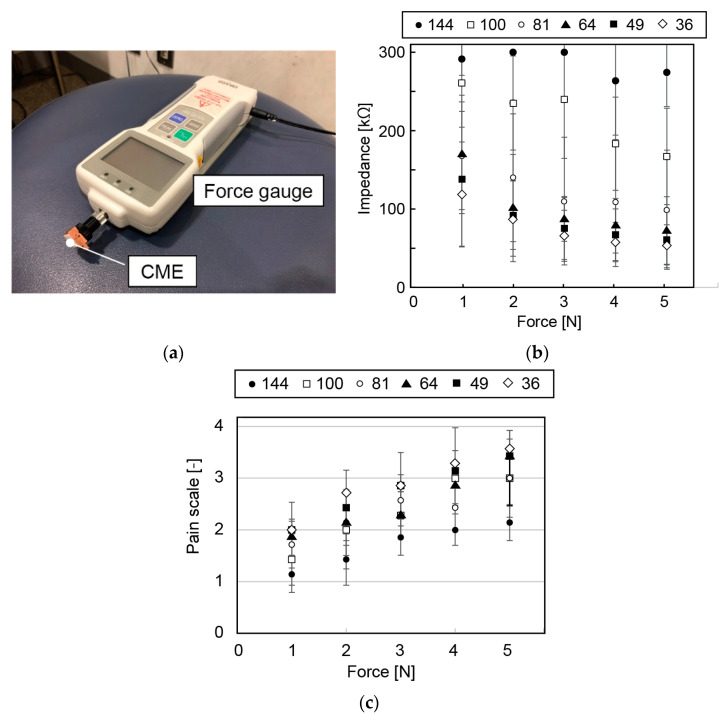
(**a**) Photograph of CME connected to force gauge. When the CME is pressed to the parietal part, the skin-to-electrode impedance is measured while the applied pressure is monitored by the force gauge. The participants recorded their perceived pain using a visual analogue scale. (**b**) Measured impedance with respect to applied force for various types of CME (summarized in Table 1). EEG can be measured when the impedance is below 300 kΩ with our experimental setup with Polymate. (**c**) Perceived pain with respect to force applied by CMEs.

**Figure 5 micromachines-14-00400-f005:**
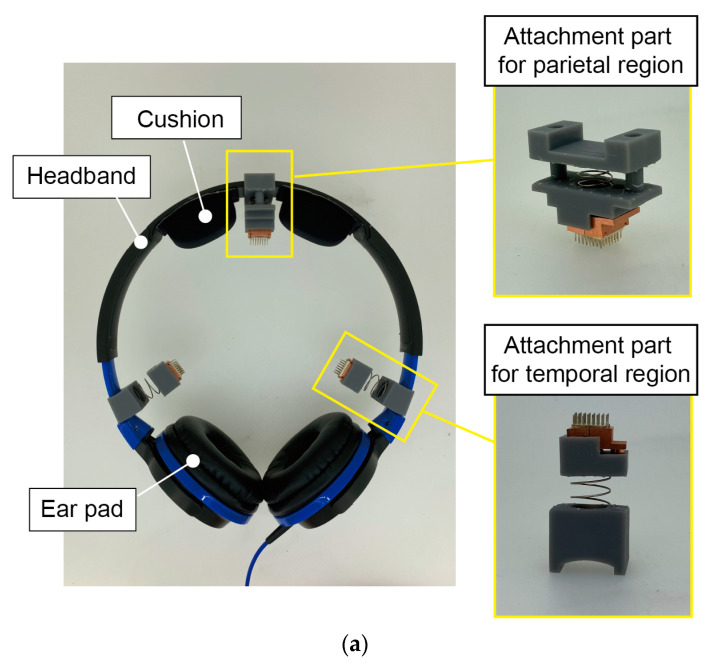
(**a**) Attachment parts of CMEs to headset. (**b**) Detailed design of attachment parts.

**Figure 6 micromachines-14-00400-f006:**
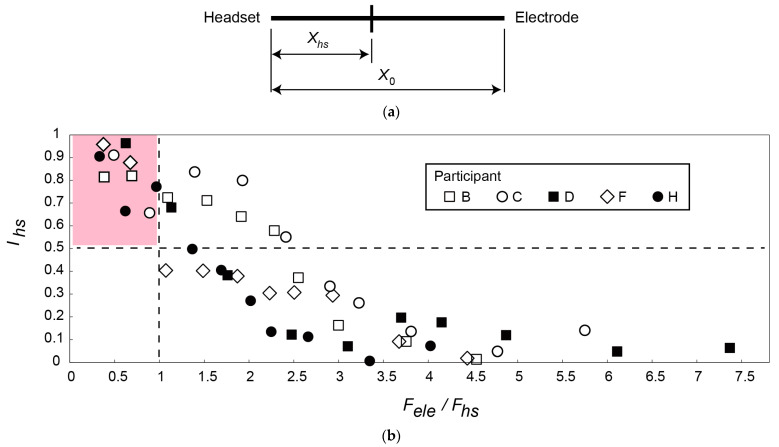
(**a**) Scale used to assess discomfort ratio. When the participants perceived pressure only from the ear pads, they made a check at the left edge. (**b**) Relationship between $F_{ele}/F_{hs}$ and headset discomfort $I_{hs}$. Each participant tested ten springs. When $F_{ele}/F_{hs}<1$, $I_{hs}$ was larger than 0.5 for all participants (i.e., they perceived the pressure applied by the ear pads to be higher than that applied by the electrodes).

**Figure 7 micromachines-14-00400-f007:**
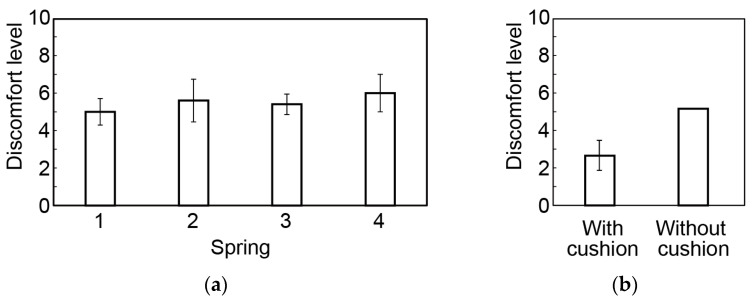
Comfort level for CMEs in parietal region. Effects of (**a**) spring stiffness and (**b**) cushion.

**Figure 8 micromachines-14-00400-f008:**
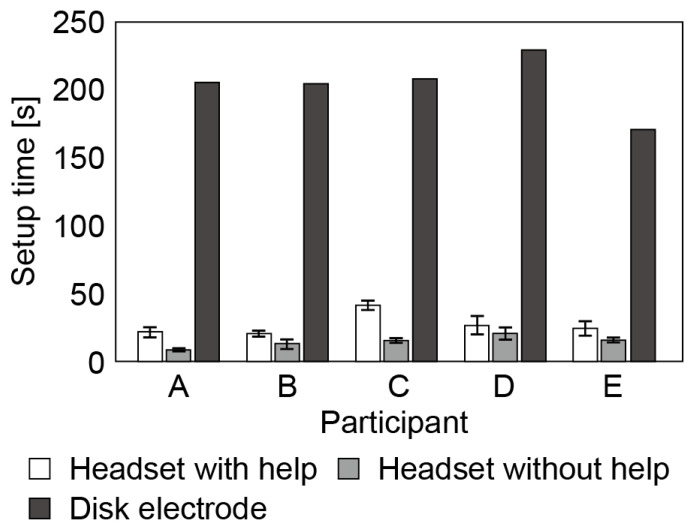
Setup time required to start measurement for headset with or without help from experimenter and for disk electrodes.

**Figure 9 micromachines-14-00400-f009:**
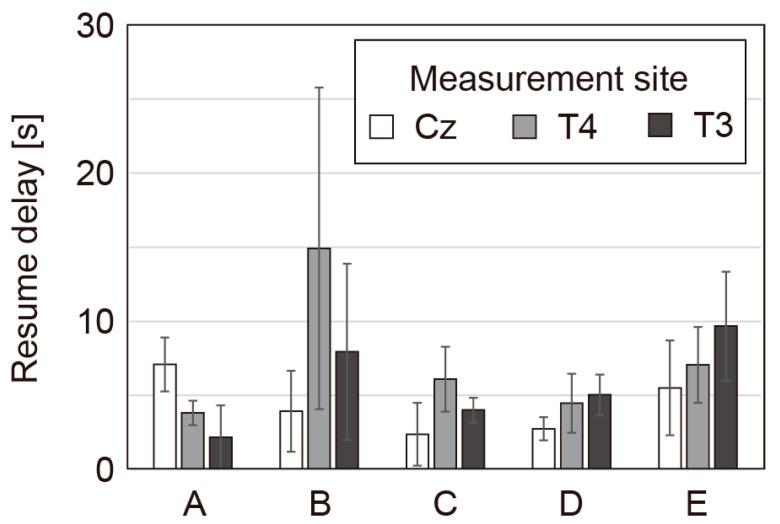
Time required to resume measurement after detachment for various measurement sites.

**Figure 10 micromachines-14-00400-f010:**
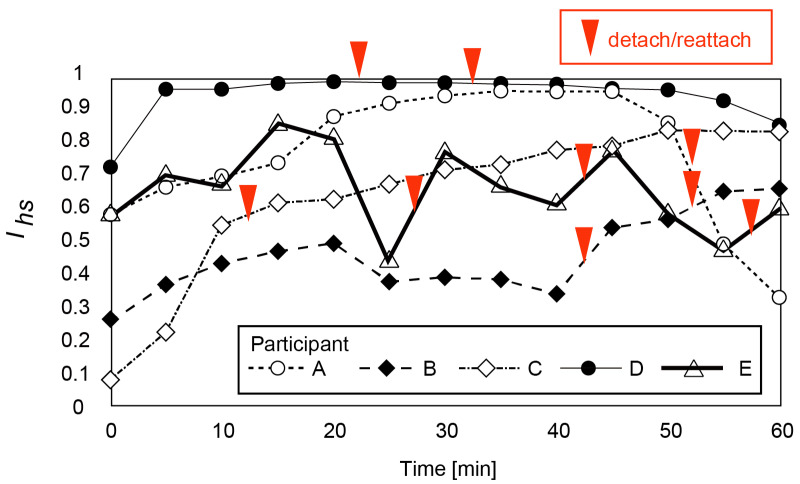
Variation of headset discomfort $I_{hs}$ for five participants. Observed detachment/reattachment times are also shown. For participants B after 40 and 50 min and participant E after 25, 40 and 55 min, detachment/reattachment of the headset resulted in an increase of $I_{hs}$.

**Figure 11 micromachines-14-00400-f011:**
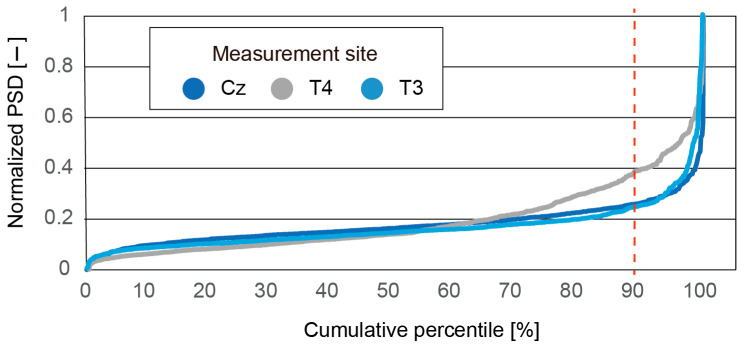
Cumulative percentile of normalized PSD of high-β/α measured at Cz, T4, and T3 for participant C. PSD is normalized by maximum value.

**Figure 12 micromachines-14-00400-f012:**
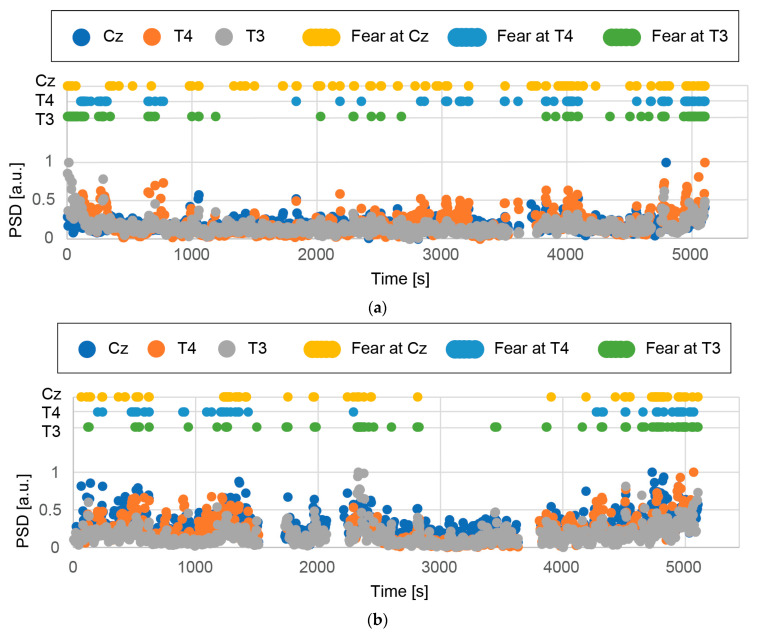
Normalized PSD for high-β/α for participants (**a**) A, (**b**) B, and (**c**) C. In the figures, the time when the PSD for high-β/αis in the top 10% is also shown with respect to the measurement sites. The analysis was conducted for periods of 5 s.

**Figure 13 micromachines-14-00400-f013:**
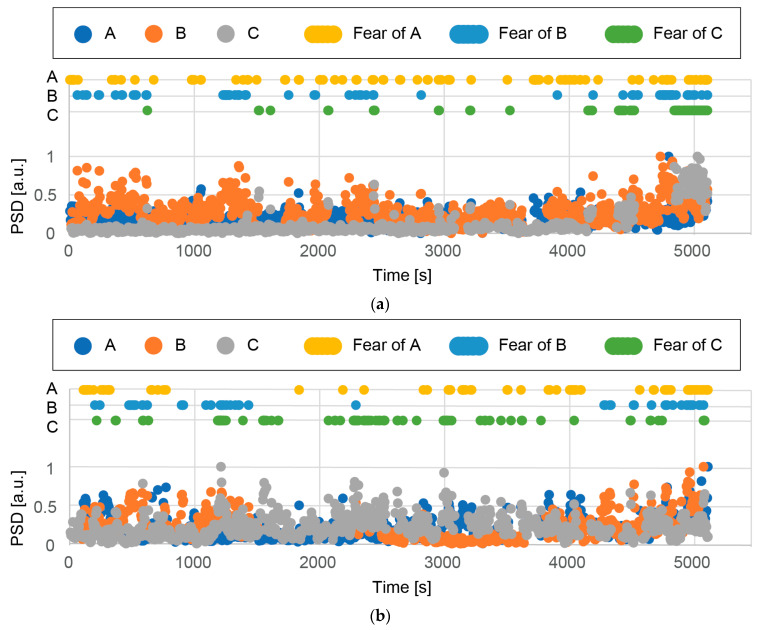
Normalized PSD for high-β/α at (**a**) Cz, (**b**) T4, and (**c**) T3 for various participants. In the figures, the time when the PSD for high-β/αis in the top 10% is also shown. The analysis was conducted for periods of 5 s.

**Figure 14 micromachines-14-00400-f014:**
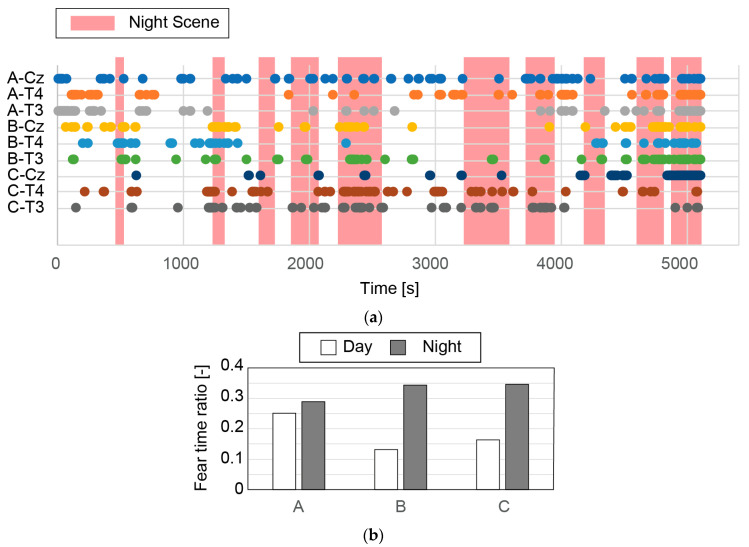
(**a**) Time when EEGs were within top 10% PSD for high-β/α for all three participants (A, B and C) and all three measurement sites (Cz, T4, and T3). (**b**) Average number of measurement sites within top 10% PSD during day and night scenes in movie for each participant.

**Table 1 micromachines-14-00400-t001:** Configuration of tested electrodes.

Number of Pillars	36	49	64	81	100	144
S [mm²]	3.9	3.1	2.5	2.0	1.7	0.34
L [mm]	2.8	2.8	2.8	2.8	2.8	0.80
w [mm]	1.4	1.1	0.89	0.73	0.60	0.43

**Table 2 micromachines-14-00400-t002:** Conditions when participants wore headset. Participants B, C, D, F, and H participated in the following experiments (bold in the table).

Participant	$l_{head-hb}$ (Left Temporal) [mm]	$l_{head-hb}$ (Right Temporal) [mm]	$l_{head-hb}$ (Top) [mm]	Opening of the Headband [mm]	$F_{hs}$ [N]
A	23.7 ± 0.12	25.0 ± 0.49	24.4 ± 0.75	190.1 ± 0.55	2.9 ± 0.12
**B**	**24.3 ± 0.31**	**29.4 ± 2.01**	**26.7 ± 3.05**	**195.7 ± 2.32**	**3.0 ± 0.06**
**C**	**29.0 ± 0.39**	**29.1 ± 0.29**	**29.0 ± 0.32**	**205.2 ± 0.31**	**3.1 ± 0.10**
**D**	**22.8 ± 0.75**	**23.7 ± 0.87**	**23.2 ± 0.87**	**219.8 ± 1.46**	**3.4 ± 0.21**
E	31.8 ± 0.88	28.9 ± 0.43	30.3 ± 1.67	201.7 ± 0.69	3.1 ± 0.21
**F**	**32.8 ± 1.40**	**34.1 ± 0.86**	**33.5 ± 1.27**	**219.8 ± 2.64**	**3.3 ± 0.11**
G	25.8 ± 0.79	25.0 ± 0.91	25.4 ± 0.85	204.4 ± 0.55	3.0 ± 0.06
**H**	**30.6 ± 0.86**	**31.1 ± 2.63**	**30.9 ± 1.78**	**209.9 ± 3.72**	**3.2 ± 0.15**
I	24.8 ± 0.71	23.4 ± 0.46	24.1 ± 0.90	207.8 ± 1.90	3.1 ± 0.06
J	28.1 ± 1.15	24.9 ± 0.81	26.5 ± 1.95	218.4 ± 3.57	3.3 ± 0.06
K	25.3 ± 0.97	23.7 ± 1.13	24.5 ± 1.29	206.9 ± 5.35	2.8 ± 0.06

**Table 3 micromachines-14-00400-t003:** Spring constant for tested springs.

Spring	1	2	3	4	5	6	7	8	9	10
Spring constant [N/mm]	0.255	0.461	0.726	1.02	1.275	1.52	1.706	2.001	2.511	3.03

**Table 4 micromachines-14-00400-t004:** Spring constant and maximum load for tested springs.

Spring	1	2	3	4
Spring constant [N/mm]	0.05	0.098	0.2	0.29
Maximum load [N]	0.35	0.75	1.2	1.8

**Table 5 micromachines-14-00400-t005:** Ratio of lost data.

Measurement Site	Cz	T3	T4
Participant A	0.19	0.25	0.22
Participant B	0.076	0.19	0.12
Participant C	0.26	0.20	0.43

## Data Availability

The data presented in this study are openly available in FigShare at https://doi.org/10.6084/m9.figshare.21819975.

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
