# Peer review of "Easily Attach/Detach Reattachable EEG Headset with Candle-like Microneedle Electrodes"

_micromachines, 2023, doi:10.3390/mi14020400_

Round 1

Reviewer 1 Report

Comments to Manuscript ID: micromachines-2191483

The manuscript presents a newly developed type of EEG electrodes which can be easily and fast placed and removed by the participant itself. Dependent on the study design this can be very helpful and advantageous. The manuscript mainly describes the development and testing of the system. Overall, the paper is well written and the study was conducted carefully.

Specific comments

Introduction line 26: ‘It reflects the neural activity of the brain...’ This would be great. However the EEG is only related to a part of the neural activity.

It was not reported how the ground and reference electrodes were mounted and removed. Kendall electrodes means one-way ECG electrodes I assume.

The system was tested with a measurement when subjects watched a horror movie. I’m sceptical if emotions can be measured reliably with the EEG. More robust methods to validate the system would either be (i) to measure an EEG with a standard laboratory system together, (ii) or to use a paradigm (e.g. acoustical oddball) which evokes well-known ERP components (e.g. N1, P2, P300). Or (iii) a combination of (i) and (ii)

Author Response

The manuscript presents a newly developed type of EEG electrodes which can be easily and fast placed and removed by the participant itself. Dependent on the study design this can be very helpful and advantageous. The manuscript mainly describes the development and testing of the system. Overall, the paper is well written and the study was conducted carefully.

Specific comments

Introduction line 26: ‘It reflects the neural activity of the brain...’ This would be great. However the EEG is only related to a part of the neural activity.

---

Thank you for your suggestion. We have modified the sentence as;

Line 26: It reflects a part of the neural activity of the brain….
---
It was not reported how the ground and reference electrodes were mounted and removed. Kendall electrodes means one-way ECG electrodes I assume.

---

KendallTM electrodes (H124SG, CardinalHealth, Japan) have wet conductive and adhesive hydrogel. You can attach them on the skin with good adhesion while you can peel them off without leaving any adhesives on the skin. We added the explanation about the electrodes as follows;

Line 153: KendallTM electrodes (H124SG, CardinalHealth, Japan) were placed attached at the right mastoid process(neutral electrode) and the left mastoid process (reference electrode). The electrodes have conductive and adhesive hydrogel and can be adhered to the skin firmly while they can also be removed without leaving any adhesives on the skin.

---
The system was tested with a measurement when subjects watched a horror movie. I’m sceptical if emotions can be measured reliably with the EEG. More robust methods to validate the system would either be (i) to measure an EEG with a standard laboratory system together, (ii) or to use a paradigm (e.g. acoustical oddball) which evokes well-known ERP components (e.g. N1, P2, P300). Or (iii) a combination of (i) and (ii)

---

Thank you for your comments. Many researchers have studied to correlate the emotion and the EEG (such as [10]), however, it is difficult to prove that the results is correct. I understand that the reviewer is skeptical about the relationship between them.

In our prior work, we compared our CME electrodes with conventional wet electrodes [28]. Good correlations and coherences were found and therefore, we consider that our CME electrodes can measure EEG as well as the conventional wet electrodes. Event related potentials (ERPs) were also successfully captured by our CMEs [29]. In this work, we assumed that emotions last for a while and did not investigate ERPs since ERPs represent immediate and short responses to given stimuli. However, as the reviewer suggests, ERPs will provide us with more information and new findings. We will investigate both ERPs and spontaneous potentials in our future work. Thank you for your precious comment.

[10] Ray, W.J.; Cole, H.W. EEG alpha activity reflects attentional demands, and beta activity reflects emotional and cognitive processes. Science 1985, 228, 750-752.

[28] Arai, M.; Kudo, Y.; Miki, N. Polymer-based candle-shaped microneedle electrodes for electroencephalography on hairy skin. Jpn J Appl Phs 2016, 55, 4–10.

[29] Yoshida, Y.; Kawana, T.; Hoshino, E.; Minagawa, Y.; Miki, N. Capturing human perceptual and cognitive activities via event-related potentials measured with candle-like dry microneedle electrodes. Micromachines 2020, 11, 556.

We have added the following sentence in Line 49.

CMEs were experimentally verified to have as good a performance as conventional flat electrodes [28] and can capture both spontaneous potential and event related potential [29].

Reviewer 2 Report

This work deals with the design of a simple device for the acquisition of EEG signals using CMEs attached to commercial audio headsets, aiming for reducing the stress and easing the installation time and procedure.

I found the work quite interesting, and its development is detailed in the work, so it is easily replicable. The extend of the study is not really very sound, as the number of cases and experiments is reduced, but it looks very prommissing.

I found only one important typo, as it makes an important paragraph incomprensible. Please review line 486 in the last paragraph in page 18.

Author Response

This work deals with the design of a simple device for the acquisition of EEG signals using CMEs attached to commercial audio headsets, aiming for reducing the stress and easing the installation time and procedure.

I found the work quite interesting, and its development is detailed in the work, so it is easily replicable. The extend of the study is not really very sound, as the number of cases and experiments is reduced, but it looks very prommissing.

I found only one important typo, as it makes an important paragraph incomprensible. Please review line 486 in the last paragraph in page 18.

We appreciate the reviewer’s encouraging words. We will keep working on this topic.

Thank you for finding the typo. We have revised the paragraph.

Line 495 (originally 486)

  1. EEGs during were successfully acquired from the three measurement sites for all three participants during the whole movie despite some periods of no data due to detachment of the CMEs and artifacts caused by body movement.

Reviewer 3 Report

The authors developed a spring electrode based on their previously reported microarray configuration. This spring electrode is claimed easily attach and reattach. But there are still several issues should be addressed.

1.      “When the flat disk electrodes are used, the high-impedance stratum corneum is removed” This statement is wired. What do the authors mean the stratum corneum is removed? Please clarify this.

2.      The authors claimed that the dry electrodes show low contact impedance but not removing the stratum corneum. This is contradictory. Please explain this.

3.      The signal noise and signal quality are highly related to the stability of electrodes and the applied pressure. In this case, the pressure is determined by the Headset. But the head shapes of different persons are variety. How to remove the various of head shapes?

4.      The authors claimed that fewer pillars cause pain. Please explain the relation between the pillar numbers and the comfort.

5.      Lower impedance lower signal-to-noise ratio. The authors also claimed that the measurement starts when impedance below 300 Ω. It is hard to understand this. Please clarify the relation between the impedance and the pillar number. The statements are contradictory in different parts.

6.      The headset system installation was complete when the impedance below 300 KΩ. This is varied from the previous claim of 300 Ω. Please explain the difference.

7.      The author analyzed the EEG obtained by the headset electrodes. It is better to compare with the gel electrodes and dry electrodes in signal-to-noise ratio and PSD.

Author Response

The authors developed a spring electrode based on their previously reported microarray configuration. This spring electrode is claimed easily attach and reattach. But there are still several issues should be addressed.

  1. “When the flat disk electrodes are used, the high-impedance stratum corneum is removed” This statement is wired. What do the authors mean the stratum corneum is removed? Please clarify this.

In order to reduce the electrode-to-skin electrodes, when the flat disk electrodes are used, the stratum corneum at the measurement site is removed, as is explained in the previous work [16, 17]. We have modified the sentence to clarify this point.

Line 33: When flat disk electrodes are used, the high-impedance stratum corneum at the measurement site is removed and conductive glue is applied to secure a conductive contact between each electrode and the skin [16,17].

[16] Teplan, M. Fundamentals of EEG measurement. Meas Sci Rev 2002, 2, 1-11.

[17] Jackson, A.F.; Bolger, D.J.; The neurophysiological bases of EEG and EEG measurement: A review for the rest of us. Psychophysiology 2014, 51, 1061-1071.

  1. The authors claimed that the dry electrodes show low contact impedance but not removing the stratum corneum. This is contradictory. Please explain this.

Dry electrodes have sharp tips and can be pressed to the skin with a large pressure to reduce the impedance even with the presence of the stratum corneum [18-25]. Our needle-electrodes can penetrate through the stratum corneum to have the low electrode-to-skin impedance. We modified the sentence as follows;

Line 37: Dry electrodes, which have pin-shaped electrodes and can be pressed to the skin with a large pressure to reduce the impedance even with the presence of the stratum corneum, have been proposed [18-25].

  1. The signal noise and signal quality are highly related to the stability of electrodes and the applied pressure. In this case, the pressure is determined by the Headset. But the head shapes of different persons are variety. How to remove the various of head shapes?

Thank you for your comment. We design the headset with springs such that the electrodes are pressed to the skin with appropriate pressure. They can compensate the differences of the head shapes to some extent. We also prepared small, medium, and large headsets to further cope with the head shapes.

Line 276: In order to satisfy , we prepared attachment parts with three sizes (large, medium, and small) to further compensate for the variation of Ihead-hs.

  1. The authors claimed that fewer pillars cause pain. Please explain the relation between the pillar numbers and the comfort.

Figure 4 (c) shows the relationship between the pillar numbers and the pain that the participant received. Discomfort decreased with the number of the pillars.

  1. Lower impedance lower signal-to-noise ratio. The authors also claimed that the measurement starts when impedance below 300 Ω. It is hard to understand this. Please clarify the relation between the impedance and the pillar number. The statements are contradictory in different parts.

First, we have to apologize our mistake. It is not 300 Ω but 300 kΩ. We used Polymate Mini (Miyuki-Giken) to record the signals. It records signals when the electrode-to-skin impedance is below 300 kΩ, as we described at Line 359. The pillar numbers determine the pressure and the space to avoid hair. The relationship between the impedance and the pillar number is described in Figure 4 (b) (the axis was revised; Ω -> kΩ). We found that the electrode with 64 pillars was the best. Thank you for your precious comment.

Line 359: The setup was considered to be complete when the impedance of each CME was less than 300 kΩ (when Polymate Mini started measuring the EEG).

  1. The headset system installation was complete when the impedance below 300 KΩ. This is varied from the previous claim of 300 Ω. Please explain the difference.

300 kΩ is the correct one. We modified them in the manuscript.

  1. The author analyzed the EEG obtained by the headset electrodes. It is better to compare with the gel electrodes and dry electrodes in signal-to-noise ratio and PSD.

In our prior work, we compared our CME electrodes with conventional wet electrodes [28]. Good correlations and coherences were found and therefore, we consider that our CME electrodes can measure EEG as well as the conventional wet electrodes. Thank you for your precious comment.

[28] Arai, M.; Kudo, Y.; Miki, N. Polymer-based candle-shaped microneedle electrodes for electroencephalography on hairy skin. Jpn J Appl Phs 2016, 55, 4–10.

We have added the following sentence in Line 49.

CMEs were experimentally verified to have as good a performance as conventional flat electrodes [28] and can capture both spontaneous potential and event related potential [29].

Round 2

Reviewer 3 Report

No further comments